computational biology/computational chemistry

motile oil droplets, adaptive behaviour, self-preservation, computational model, artificial chemistry, viability

**Author for correspondence:**
Matthew Egbert
e-mail: m.egbert@auckland.ac.nz

This article has been edited by the Royal Society of Chemistry, including the commissioning, peer review process and editorial aspects up to the point of acceptance.

# Self-preserving mechanisms in motile oil droplets: a computational model of abiological self-preservation

Matthew Egbert[1,2]

[1]University of Auckland, Auckland, New Zealand
[2]Te Ao Mārama, University of Auckland, New Zealand

ME, 0000-0002-4560-7311

Recent empirical work has characterized *motile oil droplets*—small, self-propelled oil droplets whose active surface chemistry moves them through their aqueous environment. Previous work has evaluated in detail the fluid dynamics underlying the motility of these droplets. This paper introduces a new computational model that is used to evaluate the behaviour of these droplets *as a form of viability-based adaptive self-preservation*, whereby (i) the mechanism of motility causes motion towards the conditions beneficial to that mechanism's persistence; and (ii) the behaviour automatically adapts to compensate when the motility mechanism's ideal operating conditions change. The model simulates a motile oil droplet as a disc that moves through a two-dimensional spatial environment containing diffusing chemicals. The concentration of reactants on its surface change by way of chemical reactions, diffusion, Marangoni flow (the equilibriation of surface tension) and exchange with the droplet's local environment. Droplet motility is a by-product of Marangoni flow, similar to the motion-producing mechanism observed in the lab. We use the model to examine how the droplet's behaviour changes when its ideal operating conditions vary.

## 1. Introduction

How early in the history of life might we expect to find a system capable of adaptive self-preservation? What was the first of our ancestors to move, or in some other way regulate how it interacted with its environment to satisfy its own needs? Behaviour is typically associated with sophisticated sensors and motors that are the result of a long period of evolution. This might suggest that the first forms of self-preserving behaviour occurred a long time after evolution began, but perhaps such a conclusion would be premature. Interestingly, a variety of

**Figure 1.** A ramified charge-transportation network (RCTN). When conductive chromium spheres are partially submerged in oil and subjected to a high voltage potential, they self-organize into dendritic structures such as that shown here. In this experiment [1], the spheres are 4 mm in diameter. Image used with permission of its creators.

abiological systems, including ramified charge-transportation networks [1], Bénard convection cells, motile oil droplets [2] and reaction/diffusion spots [3,4], demonstrate forms of 'self-preservation', i.e. they regulate their interaction with their environment in a way that prolongs their operation (discussed in detail in §1.1). The existence of primitive abiological self-preservation could change the way that we consider the origins of life and very earliest stages of its evolution. To elaborate: the first stages of life are seen as active only in their ability to grow or copy themselves. To this picture, we might add the possibility that the earliest forms of (pre-)life were already capable of basic self-preserving behaviours and that these behaviours might have facilitated the development of increasingly sophisticated forms of life [5]. But this notion of simple abiological systems being capable of adaptive self-preservation is unusual and counterintuitive. The purpose of this paper is thus to explain and investigate these ideas; to consider how abiological systems might be considered to have a degree of 'health' or 'viability' and how they might be able to respond in an adaptive way to regulate their environment in a way that responds to their health.

To this end, the main body of the paper presents a computational model of motile oil droplets similar to systems that have been fabricated and investigated in the laboratory (e.g. [6,7]). Previous computational models have investigated the fluid dynamics underlying the motility of these droplets [8]. We use our model to investigate how the chemical reactions taking place on the surface of droplet cause the droplet to move towards conditions that facilitate or extend the life of those very same reactions—a basic form of self-preservation. But before presenting the model, the remainder of the introduction provides context for interpreting it. In particular, we explain what we mean when we say that a system is (or is not) accomplishing an adaptive form of self-preservation.

## 1.1. Abiological systems that adapt their behaviour to accommodate their own existential needs

We have claimed that a variety of abiological systems, including ramified charge-transportation networks [1], motile oil droplets [2] and reaction/diffusion spots [3,4] accomplish basic forms of self-preservation. To justify this claim, we can start by observing that each of these systems is a low-entropy, far-from-equilibrium 'dissipative structure' that requires a steady input of energy to persist [9].

Ramified charge-transportation networks, for instance, are tree-like structures that self-organize when an electrical voltage is applied to conductive spheres that are partially submerged in oil (figure 1). The shapes that form are low-entropy structures that require the presence of electrical voltage both to form and to persist in the presence of small perturbations.

Reaction/diffusion 'spots' (figure 2) are patterns that form in the Gray–Scott model [11,12] which describes the dynamics of two diffusing chemicals ($U$ and $V$) which interact via an autocatalytic

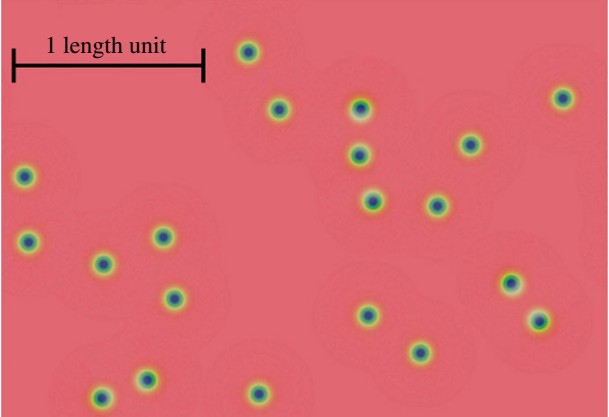

**Figure 2.** Reaction/diffusion spots. An autocatalytic reaction with diffusing reactants produces remarkable spatio-temporal patterns including these 'spots'—regions of high concentration of autocatalyst that rely upon a steady input of their 'food' reactant to persist. Reaction/diffusion spots demonstrate a range of simple, but life like behaviours including chemotaxis to high concentrations of the 'food' they require to persist and division. Image is from [10] with a scale bar added to indicate the size of 1 dimensionless length unit.

reaction whereby autocatalytic chemical ($V$) transforms precursor ($U$) into more $V$, according to the following reaction: $2V + U \rightarrow 3V$. In a range of parameters of this model, individual 'spots' form. These are regions of high autocatalyst concentration that are also dissipative structures, i.e. low-entropy configurations of material that are inherently unstable, but that can persist when the system is 'fed' or 'driven' by a source of energy.

Both the reaction/diffusion spots and the ramified charge-transportation networks demonstrate move in a way that positively contributes to the processes that create them—a basic form of self-preservation. Specifically, the ramified charge-transportation networks accomplish 'energy-seeking behaviour' [1]: they require high voltage potentials to persist and they reorient themselves to span high voltage potentials. In so doing, they are amplifying the process that creates and stabilizes them. Similarly, the energy that sustains the reaction/diffusion dissipative structures (the 'spots') is provided by a 'food' chemical ($U$) and these structures automatically (i.e. without any added mechanism) move towards local regions that are higher in concentration of $U$ [4].

Each of these systems has what might be labelled *existential needs*: conditions that must be met for it to persist. Each system also appears to behave in ways that satisfy those needs. Non-dissipative structures can also be said to have existential needs (e.g. a rock must not exceed its melting temperature if it is to remain a rock) but the needs of dissipative structures are different from the existential needs of entities like rocks in an important way. The difference lies in the fact that the way these two classes of systems exist is fundamentally different: rocks (and other non-dissipative entities) are merely passively stable, whereas dissipative structures are constantly falling apart and yet persist thanks to processes of repair, replacement or reconstruction [9,13]. This means that existence for passively stable entities is the absence of a destructive event. By contrast, for dissipative structures, existing is a process—and a process that must continue for the system to persist. Processes have rates and as such it is possible to measure or quantify or respond to how 'viable' a dissipative structure is—i.e. how well it is doing at persisting despite its intrinsic tendency to degrade (cease to exist) [14]. There is no equivalent measurement for passively stable systems as their existence is not a process in the same way that it is for dissipative structures.

Being able to measure or quantify the viability of a dissipative structure becomes particularly interesting when we consider the possibility of a dissipative structure that can 'monitor' or change its behaviour in response to *its own* viability dynamics. We can find examples of this kind of 'viability-based' behaviour in modern organisms; for instance in the metabolism-dependent chemotaxis modern bacteria such as *Escherichia coli* [15] and *Azospirillum brasilense* [16]. These bacteria move towards certain attractants not by sensing them directly, but by responding to how well their metabolism is operating. Instead of sensing and responding to what is in their the environment (to anthropomorphize: 'the more I travel in this direction, the more food there is, so I'll keep heading this way'), metabolism-dependent behaviour senses and responds to the metabolic-efficacy, (e.g. by responding to the state of their electron transport system [15,17]) 'the more I travel in this direction, the more quickly my metabolism is operating, so I'll keep heading this way'.

A simple way to distinguish between behaviours that are viability-based and those that are not is to consider what happens when needs of the organism change. This is, in fact, one of the ways that researchers have determined empirically whether a particular form of chemotaxis is metabolism-dependent or metabolism-independent. If mutants that have lost the ability to metabolize a particular attractant continue to move towards that reactant, the chemotactic mechanism must be metabolism-independent as that attractant no longer has any effect upon metabolism [18, p. 1590].

By contrast, when a behaviour is metabolism-dependent (or more broadly construed, 'viability-based'), organisms can respond appropriately to phenomena neither they nor their ancestors have ever previously encountered [19,20]; integrate the combined impact of diverse influences upon their metabolic state without dedicated machinery [19,21]; adapt to changes in their own internal operation [20], including changes that affect their own abilities and needs [19,22]; and can facilitate evolutionary dynamics by transforming what would otherwise be a detrimental mutation into one that is beneficial [5].

With these ideas in place, we can now be explicit about what we mean by adaptive self-preserving behaviour. For a system to be considered to accomplish self-preserving behaviour in the sense used in this paper, it must

1. be a dissipative structure,
2. modify its interaction with its environment (behaviour),
3. do so in a way that tends to prolong the existence of the dissipative structure (self-preserving),

if the following additional criterion is satisfied,

4. the behaviour must be driven by a response to the dissipative structure's viability dynamics (how well it is doing at persisting)

then the behaviour is adaptive and 'viability-based.' Behaviours can be adaptive without being viability-based—e.g. the metabolism-*independent* chemotaxis of bacteria can adapt to different environmental conditions, but does so in a way that does not directly respond to the bacterium's viability dynamics.

Viability-based adaptive self-preservation is particularly interesting in the context of the origins of life as (i) it provides a range of advantages in terms of robustness and survival (several of which are listed above and discussed in [5,19,20]); and (ii) is simply implemented—so simply that it is found in simple dissipative structures. If present at the earliest stages of life's evolution, viability-based behaviour could have played an important role, using its benefits to facilitate life's emergence and early development, bringing us back to the questions that we opened with: How early in the history of life might we expect to find a system capable of adaptive self-preservation? What was the first of our ancestors to move, or in some other way regulate the way that it interacted with its environment in response to its own needs? Were there pre-biotic dissipative structures that were already capable of changing their behaviour in response to their own viability? For the remainder of this paper, we focus on just one such system: motile oil droplets. Using a computational model, we investigate how these systems are indeed capable of changing their behaviour in a viability-based response to their own existential needs. To be precise, it is not the oil droplet itself that is self-preserving. Droplets of oil in water are passively stable entities and not dissipative structures. By criterion (1) above, this discounts the oil droplet as a candidate self-preserving system. Instead, we find in our model a self-preserving system in the combination of (i) the chemistry on the surface of the droplet; (ii) the marangoni flow that it creates; and (iii) motility driven by the marangoni flow. These three processes support one another, resulting in a precarious dissipative system that moves itself towards local environmental conditions that are beneficial to its dynamic viability—a basic form of viability-based adaptive self-preservation.

The remainder of the paper proceeds as follows. The next section describes one kind of motile oil droplet and its mechanism of motility and why it is an interesting potential example of a dissipative structure that accomplishes adaptive self-preservation. The subsequent sections present a new computational model that we use to investigate in more detail how motile oil droplets accomplish viability-based behaviour and how that behaviour automatically adapts when the system's existential needs change.

## 2. A model of motile oil droplets

Researchers interested in the origin of life have investigated motile oil droplet (MOD) systems, where the oil droplet's movement towards resources and away from waste products has been suggested to play an

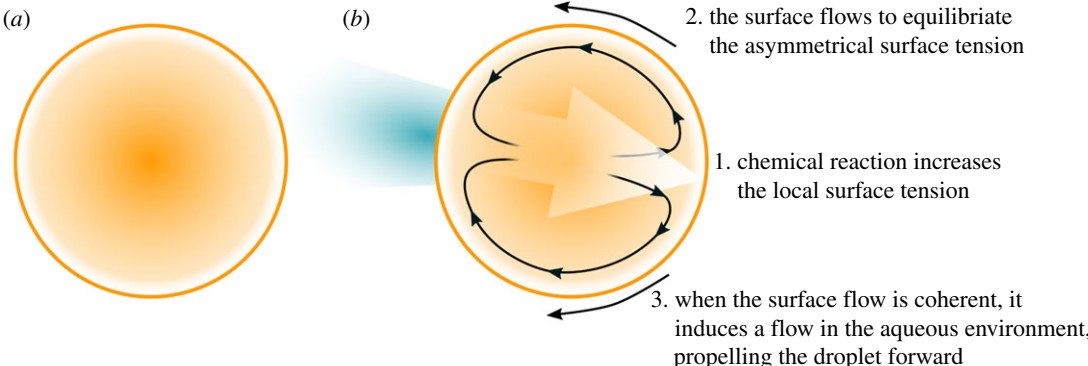

**Figure 3.** An initially symmetrical droplet breaks symmetry (*a*) and motion producing convection cells form (*b*). Copied from [7].

Annotations in figure:
2. the surface flows to equilibrate the asymmetrical surface tension
1. chemical reaction increases the local surface tension
3. when the surface flow is coherent, it induces a flow in the aqueous environment, propelling the droplet forward

important role in maintaining their far-from-equilibrium state [7]. In one such experiment, a nitrobenzene oil droplet containing oleic anhydride is placed in a high pH aqueous environment with a surfactant that facilitates the formation of droplets. On the surface of the droplet, the oleic anhydride is hydrolysed into two amphiphilic molecules that change the local surface tension. The surface flows to equilibrate any asymmetry in the surface tension and when this Marangoni flow is coherent, the flow at the surface induces a flow in the local aqueous environment via fluid friction, propelling the droplet through the aqueous medium (figure 3).

Wherever the hydrolysis of the precursor (henceforth 'the reaction') operates more quickly, more surfactant is produced. The Marangoni instability causes surface flow away from these regions, and this means that the droplet's motion tends to be towards the environmental conditions where the reaction is happening the most quickly. For instance, when placed in a pH gradient, the droplet moves up the pH conditions, because these conditions are more conducive for the motion-generating reaction to take place [7]. The reaction moves itself towards the conditions it needs to persist, and in this way is acting to satisfy its own needs, in what has been called a basic form of autonomous agency or cognition [6]. The model we present here allows us to examine this claim in more detail and to investigate the limits and capabilities of such behaviour. It simulates a two-dimensional space that contains a single motile oil droplet (MOD) and its environment of diffusing chemicals. We now describe each of these elements of the model (figure 4).

## 2.1. Environment

The simulated environment is a circular 'Petri dish', 10 mm in radius. It contains one motile oil droplet and two diffusing 'environmental chemicals' ($H_E$ and $P_E$), which we distinguish from 'surface chemicals' ($H_S$, $P_S$ and $A_S$) that are embedded in the droplet's surface (described below). The concentration of each environmental chemical is a function of position in a two-dimensional space ($\mathbf{p} \in \mathbb{R}^2$) and time ($t \in \mathbb{R}$), which changes by way of chemical diffusion and exchange with the surface of the MOD according to the following differential equations:

$$\frac{\mathrm{d}P_E(\mathbf{p}, t)}{\mathrm{d}t} = D_P \nabla^2 P_E \; - \chi_P(\alpha_P P_S - (1 - \alpha_P)P_E^*) \tag{2.1}$$

and

$$\frac{\mathrm{d}H_E(\mathbf{p}, t)}{\mathrm{d}t} = \underbrace{D_H \nabla^2 H_E}_{\text{diffusion}} - \underbrace{\chi_H(\alpha_H H_S - (1 - \alpha_H)H_E^*)}_{\text{surface/env. exchange}}, \tag{2.2}$$

where $D_v$ is the diffusion constant of $v$ and the last term in each equation is a function that specifies the exchange between the MOD surface and the environment (described in detail below).

## 2.2. Motile oil droplets

The MOD is modelled as a disc of fixed radius $r = 1$ mm. Its centre, $\mathbf{q} \in \mathbb{R}^2$, is constrained such that the entire MOD always lies within the Petri dish. We simulate the dynamics of the MOD's surface chemistry, involving three 'surface chemicals' or 'reactants': $H_S$, $P_S$ and $A_S$. The concentration of each reactant is a

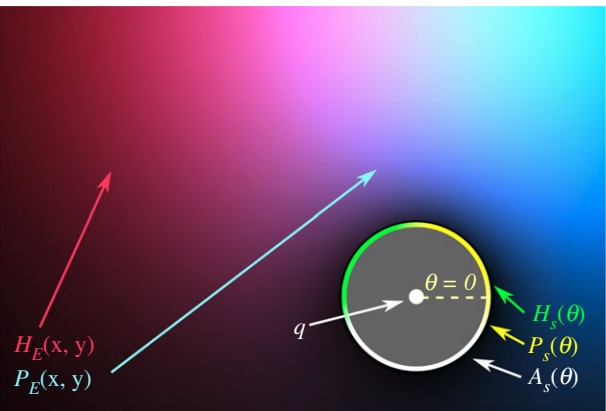

**Figure 4.** Schematic indicating the key variables in the model. Environmental concentrations of reactants ($H_E$ and $P_E$) are distributed in a two-dimensional environment, while the concentration of the surface reactants ($H_S$, $P_S$ and $A_S$) are a function of their position on the surface of the motile oil droplet ($\theta$).

**Table 1.** Model parameters including diffusion rates; surface tension constants; the propensities for each reactant to enter the MOD interface (lower values describe more amphiphilic molecules, which prefer the surface of the MOD to the aqueous environment); and the rate at which reactants transition between the MOD surface and the aqueous environment.

| diffusion rate constants | surface tension constants | interface preference constants | surface/environment exchange rates |
|---|---|---|---|
| $D_P$ : 0.5 | $\gamma_P$ : 0 | $\alpha_P$ : 0 | $\chi_P$ : 1.25 |
| $D_H$ : 0.05 | $\gamma_H$ : 0 | $\alpha_H$ : 0.1 | $\chi_H$ : 5 |
| $D_A$ : 0.5 | $\gamma_A$ : 50 | n.a. | n.a. |

function of its position on the surface of the MOD, ($\theta \in [0, 2\pi)$) and time. Its dynamics are defined by the linear sum of several processes as described by the following differential equations:

$$\frac{\mathrm{d}A_S(\theta, t)}{\mathrm{d}t} = 2kPH + D_A \frac{\partial^2 A_S}{\partial \theta} + K_\Gamma \frac{\partial \Gamma}{\partial \theta} A_S \tag{2.3}$$

$$\frac{\mathrm{d}P_S(\theta, t)}{\mathrm{d}t} = -kPH + D_P \frac{\partial^2 P_S}{\partial \theta} + K_\Gamma \frac{\partial \Gamma}{\partial \theta} P_S + \chi_P(\alpha_P P_S - (1 - \alpha_P)P_E^*) \tag{2.4}$$

and

$$\frac{\mathrm{d}H_S(\theta, t)}{\mathrm{d}t} = \underbrace{-kPH}_{\text{reaction}} + \underbrace{D_H \frac{\partial^2 H_S}{\partial \theta}}_{\text{diffusion}} + \underbrace{K_\Gamma \frac{\partial \Gamma}{\partial \theta} H_S}_{\text{Marangoni flow}} + \underbrace{\chi_H(\alpha_H H_S - (1 - \alpha_H)H_E^*)}_{\text{surface/env.exchange}} \tag{2.5}$$

which we now describe.

The first term of each of equations (2.3)–(2.5) describes the change in surface reactants caused by an autocatalytic chemical reaction whereby $A$, an amphiphilic molecule that increases surface tension, $H$ which is a proxy for alkalinity, and $P$ a precursor molecule that is transformed into $2A$ when $H$ is present according to the following reaction,

$$P + H \rightarrow 2A, \tag{2.6}$$

which has a forward reaction rate of $k = 1 \times 10^4$.

The second term describes the diffusion of the reactants around the surface of the MOD. The third term describes the equilbriation of surface tension (Marangoni flow). To calculate this, each species is associated with a surface tension constant, $\gamma$ (table 1). The interfacial tension, which varies locally over the surface of the droplet is assumed to be proportional to the sum of the local concentration of the species, multiplied by their surface tension constant.

$$\Gamma(\theta, t) = \gamma_A A_S + \gamma_P P_S + \gamma_H H_S, \tag{2.7}$$

but given that $\gamma_P = \gamma_H = 0$, this equation simplifies to $\Gamma(\theta, t) = \gamma_A A_S$.

Asymmetry in $\Gamma$ is resolved by a flux of all reactants, $G$, proportional to the gradient of $\Gamma$.

$$G = -K_\Gamma \frac{\partial \Gamma}{\partial \theta}, \tag{2.8}$$

which translates to the third term in each of equations (2.3)–(2.5), which states that the concentration of each reactant changes at a rate proportional to its local concentration and the velocity of the Marangoni induced flux. This term is similar to that of diffusion, but note that the gradient that is being relaxed is that of total surface tension, not the concentration of the species. Thus, unlike diffusion, if this process were the only one operating, it could maintain an asymmetrical distribution of chemicals (given asymmetrical initial conditions). The scaling factor $K_\Gamma = 1$ was chosen to have diffusion and Marangoni flow have comparable scale influence upon reactant distribution. Diffusion and Marangoni flow are simulated using the forward time, centred space (FTCS) numerical scheme, where the surface of the droplet is represented by 48 discretizations, and the environmental concentrations are simulated over a $128 \times 128$ lattice.

The last terms of equations (2.1), (2.2), (2.4) and (2.5) describe exchange of reactants between the MOD surface and the environment. This exchange approaches an equilibrium which is defined by two reactant-specific parameters: $\alpha \in [0, 1]$, which specifies the tendency of the reactant to enter the MOD interface; and $\chi$, which specifies the rate at which equilibrium is approached. Lower values of $\alpha$ describe chemicals that are more strongly amphiphilic, i.e. whose equilibrium state has a higher concentration in the interface; higher values of $\alpha$ describe chemicals that are more hydrophilic i.e. whose equilibrium concentrations involve fewer (or no) reactants embedded in the interface. In these terms, $A_E^*$ and $H_E^*$ represent the *nearby* environmental concentrations of $A$ and $H$. To calculate these values, a weighted sum of local environmental concentrations is taken,

$$A_E^* = A_E \odot \omega_{(\mathbf{q},\theta)} \tag{2.9}$$

and

$$H_E^* = H_E \odot \omega_{(\mathbf{q},\theta)}, \tag{2.10}$$

where $\odot$ is elementwise multiplication, and $\omega_{(\mathbf{q},\theta)}$ is a weighting kernel used to sample the portion of the MOD's environment that is close to the position $\theta$ on its surface. What qualifies as nearby is different for different positions around the perimeter of the MOD and so each discretization has its own weighting kernel, which is a function of the position of the MOD ($\mathbf{q}$) and the angular position of the discretization ($\theta$). Each kernel has the same lattice size ($128 \times 128$ elements) as that used to simulate the environment and the value given to each element is defined by the following equation:

$$\omega_{\mathbf{q},\theta}(i, j) = \begin{cases} 0: & \text{if } \sqrt{x^2 + y^2} > r \\ \exp\left(\frac{-16}{\pi^2} \left| \theta - \tan^{-1}\left(\frac{y}{x}\right) \right|^2 \right): & \text{otherwise,} \end{cases} \tag{2.11}$$

where $r$ is the MOD's radius and $i, j$ are indices of the kernel matrix. To explain this equation: each kernel element is associated with a position in the simulation, and we use $x$ and $y$ to describe that position relative to the centre of the MOD thus: $x = (q_x + R) - i\Delta X$ and $y = (q_y + R) - j\Delta X$, where $R = 10$ mm is half the width of the simulated environment and $\Delta X = 2R/128$.

With these values defined, the piecewise function gives 0 weighting to any elements that are outside of the droplet's circle. Within its circle, elements are given higher weighting when their angle relative to the centre of the MOD ($\tan^{-1}(y/x)$) is closer to the $\theta$ of the relevant surface discretization. The weighting of elements falls off according to a Gaussian function of the difference of this angular distance. Finally, each weighting kernel is normalized such that the sum of all of its elements is 1. Visualizations showing kernels for two different regions of the MOD's surface are provided in figure 5.

The final aspect of the model to explain is the motion of the MOD. Marangoni flow induces a tangential flow in the surrounding medium, resulting in an acceleration. We assume viscosity to be high and model velocity as proportional to this acceleration, thus:

$$\frac{\mathrm{d}\mathbf{q}}{\mathrm{d}t} = k_v \int G(\theta)(\theta + \pi/2) \, \mathrm{d}\theta, \tag{2.12}$$

where $k_v = 2.5 \times 10^4$ is a constant that scales the amount of motion produced by the Marangoni flux, and the term $(\theta + \pi/2)$ specifies that the force applied to the MOD is tangential to the surface.

convolution kernels ($\omega_{(\vec{q}, \theta)}$)

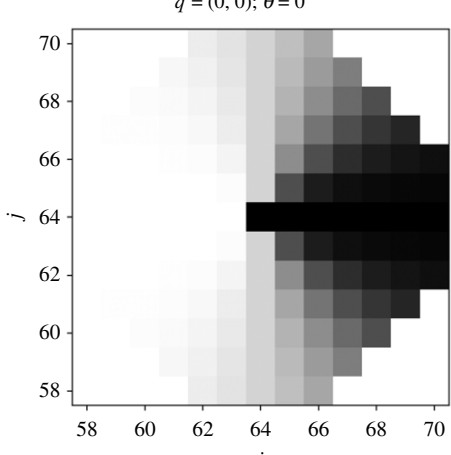

$\vec{q} = (0, 0); \theta = 0$

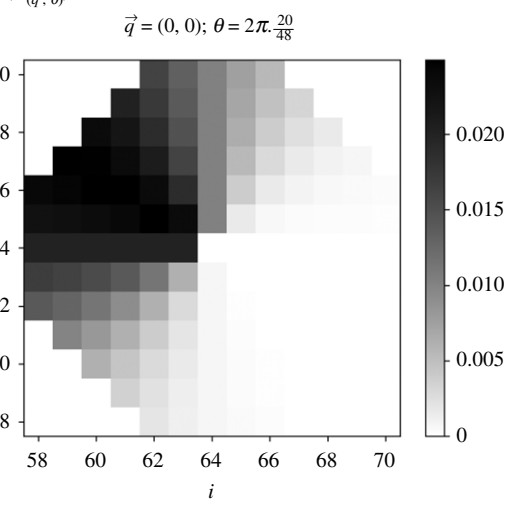

$\vec{q} = (0, 0); \theta = 2\pi.\frac{20}{48}$

**Figure 5.** Each discretization around the surface of the MOD has its own weighting kernel which is used to calculate its local environmental reactant concentrations. Here, portions of two of these kernels are plotted both for an MOD placed at the centre of the simulated Petri dish to show how different parts of the MOD's surface interact with different portions of the environment. The left plot shows the kernel for the 'eastern' portion of the MOD's surface ($\theta = 0$) while the right plot shows the kernel for the part of the MOD's surface that faces roughly 'northwest' ($\theta = 2\pi\frac{20}{48}$). All elements not plotted have a value of 0.

## 3. Experiments and results

We now present a series of three experiments that use this model. In the first, we recreate the droplet chemotaxis results observed in laboratory experiments. In the second, we consider the possibility of droplets similar to those just described, but now also capable of refuelling themselves by picking up precursor from their environment. Finally, the third experiment varies the rates at which the precursor reactant is adsorbed from the environment. We use this final experiment to investigate the extent to which the behaviour can adapt to changes in its own existential needs.

### 3.1. Experiment 1: chemotaxis

We start our exploration of this model by recreating the qualitative results of the empirical MOD work described in §2 and published in [7]. The MOD is seeded with uniform surface concentrations $A_S = P_S = 10^{-4}$, $H_S = 0$ and placed in an environment with a two-dimensional Gaussian distribution of $H_E$, with a maximum concentration of $H_E^{\max} = 0.002$ located at ($-1.5$ mm, 0 mm) (where the origin is the centre of the petri dish) with a standard deviation of $2\frac{1}{2}$. For simplicity, in this first experiment, we artificially clamp the environment such that its state does not change from its initial condition throughout the trial (i.e. diffusion does not take place in the environment, and the adsorption of reactants from the environment onto the droplet has no effect upon the environmental concentration). In later experiments we allow the environment to change.

The MOD's initial position is offset from the peak of the gradient at $\mathbf{q} = (0$ mm, $-4$ mm), and thus its local environmental distribution of $H_E$ is asymmetrical. This results in an asymmetric production of $A_S$, with regions of the droplet's interface that are situated in an environment higher in $H_E$ producing proportionally more $A_S$. This, in turn, produces an asymmetrical surface tension, which causes Marangoni flow from the up-gradient portions of the interface to those with lower concentrations of $A_S$. In a way that is comparable to the motility mechanism described in [7], the Marangoni flow causes the droplet to move towards the conditions where the motion-producing reaction (equation (2.6)) is proceeding more rapidly.

Figure 6 presents a snapshot shortly after the start of this experiment, indicating the asymmetrical distribution of the environmental 'resource' ($H_S$) and reaction product ($A_S$) at the leading surface of the droplet ($\theta \approx \pi/2$), and the higher amount of precursor ($P_S$) at the tail end.

After some time ($t \approx 15$), the precursor is depleted, the diffusive and Marangoni processes resolve the interfacial tension disequilibrium, and the MOD stops moving before reaching the peak of the resource,

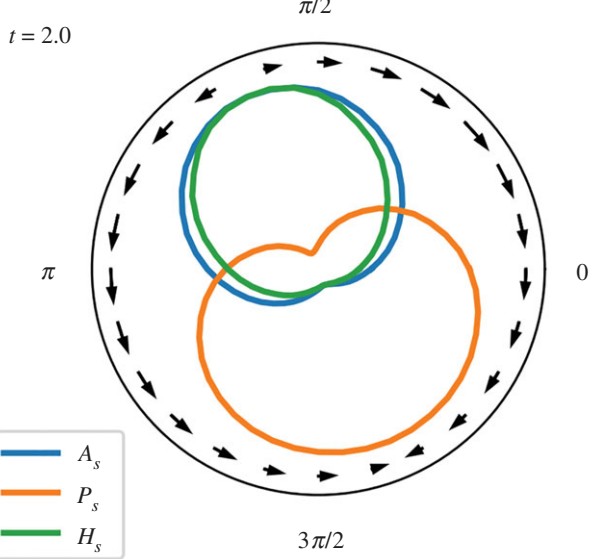

**Figure 6.** Snapshot from Experiment 1A showing surface-reactant distribution and Marangoni flow shortly after the start of the experiment ($t = 2$). The farther the reactant line is from the centre of the plot, the higher its concentration in that region of the MOD's surface. Peripheral arrows show direction and magnitude of Marangoni flow which is working to equilibrate the asymmetrical surface tension, which is the result of the asymmetrical distribution of $A$. The centre figure in the top of figure 7 indicates the position of the MOD relative to the gradient as well as its motion.

as depicted in figure 7a. Once created there is no degradation or removal of $A_S$ and so it accumulates in the surface of the drop during the course of the simulation.

If we prevent the droplet from moving, then it remains in an environment lower in $H_E$, and the precursor is more slowly transformed into $A_S$ (figure 7b). The asymmetrical surface tension ($\Delta\Gamma_{\max}$) is maintained for a longer time as the reaction depletes $P_S$ less quickly.

If we remove the artificial clamping of the environment, two processes now influence the environmental concentration of the $H_E$: the droplet now adsorbs $H$ from its immediate surroundings, and the reactants in the environment diffuse, partially compensating for the local depletion of $H_E$. For both the motile and non-motile droplet, the primary result appears to be a reduction in the reaction rate, and a longer period of time passes before the precursor is depleted.

In all cases, the total amount of $A_S$ produced is the same, which as might be expected, is equal to its initial concentration plus two times the initial concentration of the precursor, $A_S^{\text{final}} = 3 \times 10^{-4} = A_S^0 + 2P_S^0$.

## 3.2. Experiment 2: refuelling chemotaxis

What happens if the droplet is capable of replenishing precursor from its environment? To explore this possibility, we add a second environmental resource gradient, this time of precursor ($P_E$). The gradient is the same as that of $H_E$, except that its centre is placed to the right of the Petri dish's centre, at (+1.5 mm, 0 mm). Now, as the droplet moves through the environment, it can pick up additional 'fuel' allowing its motion-producing reaction to continue.

Figure 8a shows this condition with the environment once again clamped to its initial state. The droplet starts as the same location as before, but moves for a longer period of time, coming to rest in between the peaks of the two gradients, slightly closer to the $P_E$-peak.

Figure 8b shows what happens when we fix the position of the MOD. The presence of $P$ in the environment means that the droplet produces $A_S$ at a constant rate. The motile droplet moves (figure 8a) to a higher concentration and thus produces $A_S$ more rapidly than the stationary droplet (figure 8b).

In figure 8c, we see what happens when we allow the environment to change. In these plots, the droplet's position is clamped and it depletes its local resources (note the black halo around the droplet in the centre plot). Resources from elsewhere in the environment diffuse into this void, and $A$-production continues, but at a slower rate than either of the fixed environment conditions (final $dA_S/dt$ for Experiment 2B was $\approx 1.228 \times 10^{-5}$ and for Experiment 2C is $\approx 7.221 \times 10^{-6}$).

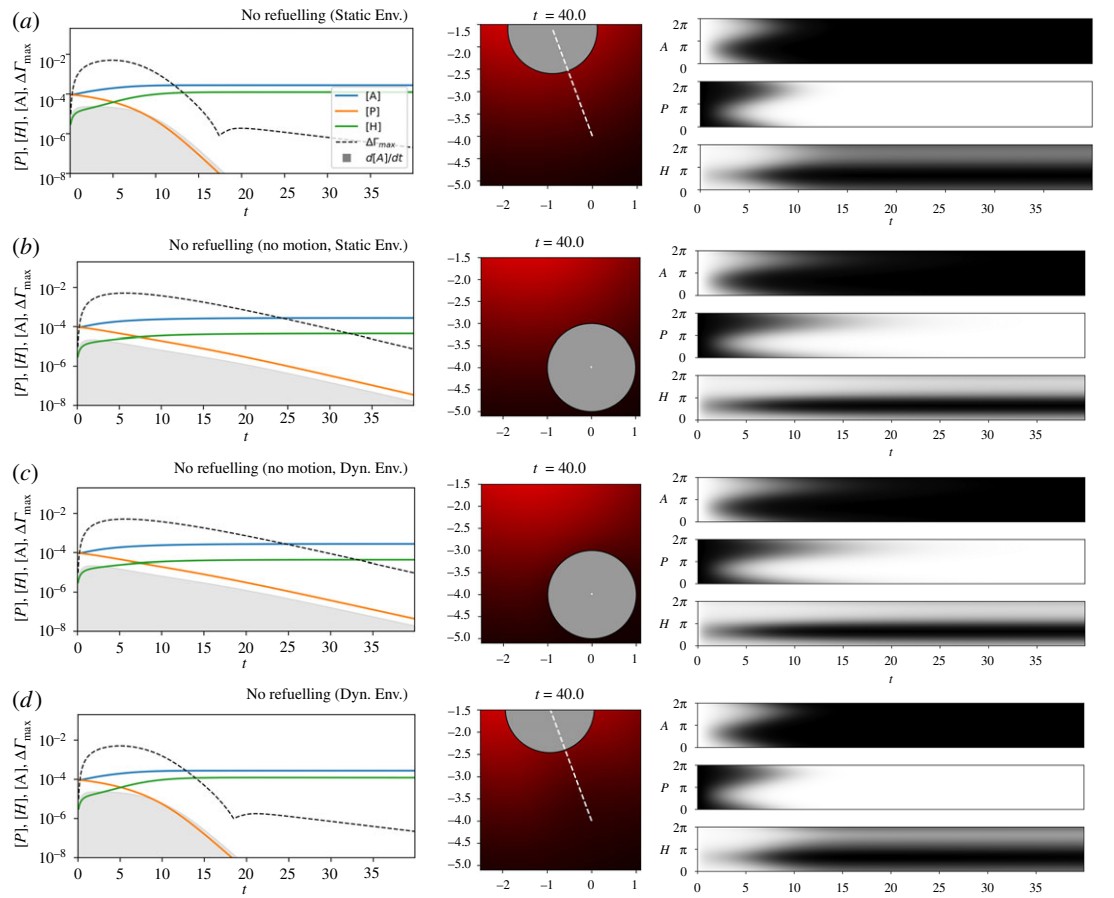

**Figure 7.** Experiment 1: MOD chemotaxis. These plots compare motile and stationary MOD in fixed and dynamic environmental conditions. On the left, time series show the average concentration of each surface-reactant; the maximum difference in surface tension between any two points on the MOD's surface (dashed line); and the rate of $A$-production (grey shaded area). The middle plots show the final state of the environment (concentrations are normalized, so colours are not directly comparable between plots) with the final position of the droplet (circle) and its trajectory (dashed white line). The right column of plots show how the (normalized) concentration of each chemical varies over the surface of the droplets (vertical axis) over time (horizontal axis).

Finally, in figure 8d, we see what happens in a dynamic environment when the droplet is allowed to move. Here the droplet consumes the resources in its local environment, but in the process moves towards environments that provide the resources needed for continued motion. The motion of the droplet in this condition involves a back-and-forth between the two resource peaks, with periods of time in which $P_E > H_E$ and vice versa. After some time, the MOD has, figuratively speaking, painted itself into a corner—at the end of the trial, it is low in $P_S$ but the pathway back to environments with $P_E$ is blocked by environments that have already been visited and are thus depleted in $P_E$. The droplet cannot 'see' distally, and is unable to make the necessary movements to keep the reaction going. Perhaps if the simulation were to run for much longer, the diffusing environment reactants would eventually provide a pathway for the droplet to resume its motion and the reaction that drives it.

Note that the peak surface tension difference ($\Delta\Gamma_{\max}$) in the left plot of figure 8a, undergoes a series of 'bounces' before eventually stabilizing. Each of these dips corresponds to a reversal of direction of the droplet. To understand what is going on here, it is useful to think about where on the surface of the droplet $A_S$ is being produced (as it is $A_S$ that changes the local surface tension). Figure 9 indicates the rate of $A_S$ production at different positions around the surface of the MOD (vertical axis) as a function of time. This value is calculated by taking the product of the local concentration of $H_S$ and $P_S$ at each time step. The surface location that is producing $A_S$ at the highest rates at each sampled time step is highlighted in yellow.

It is apparent from examining this figure that the rate of maximal production is switching back and forth from one side of the MOD to the other. We can explain this motion in the following way. Consider a

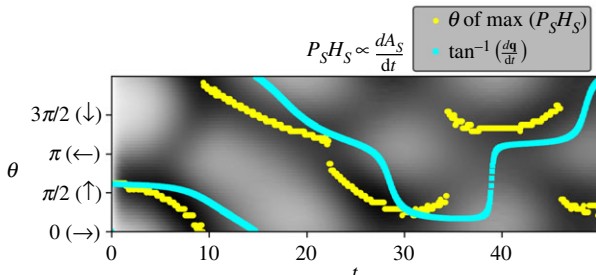

**Figure 8.** Experiment 2, where a second environmental resource allows the MOD to 'refuel' by acquiring additional precursor. The caption for figure 7 explains how to interpret these figures.

**Figure 9.** Grey shading shows the rate of $A_S$-production for different locations on the surface of the droplet as a function of time. These values are approximated by measuring the product of the concentrations of $P_S$ and $H_S$ around the surface. The local maximal rates of $A_S$ production (yellow) around the surface of the droplet predict the near-future direction of travel (cyan). Data taken from the refuelling static environment experiment (Experiment 2A).

moment in the simulation when the MOD is closer to the $P_E$ peak ($t \approx 6.5$). In this situation, $P$ is relatively abundant, and we would thus expect the availability of $H$ to be the limiting factor for the reaction rate. The left ($\theta = \pi$) side of the MOD would be higher in $H_E$ than the right ($\theta = 0$) and we would thus expect the overall production rate of $A_S$ to be higher on the left, driving motion in that direction. These details are confirmed when we examine the concentrations at these locations (figure 10). $P$ is significantly higher at $\theta = 0$ but the overall production rate is higher at $\theta = \pi$, where $H_E$ is higher.

When closer to the $H_E$ peak (e.g. at $t \approx 9.5$), the limiting factor becomes $P$ and the side of the MOD which is higher up the $P_E$ gradient (i.e. the right-hand side, where $\theta = 0$) is now the side that is producing $A_S$ more quickly (figure 11). As made clear in the next experiment, the two scenarios

At $q_x$-maximum ($t \approx 19.2$)

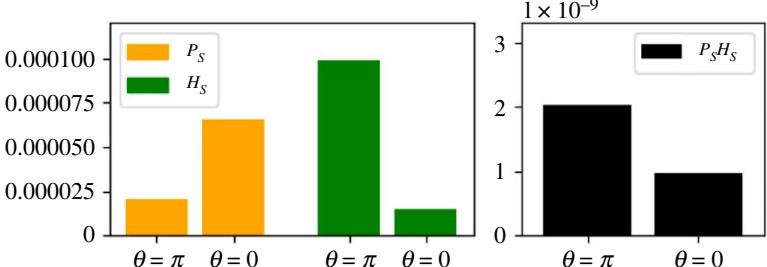

**Figure 10.** The concentration of $P_S$ and $H_S$ at $t \approx 6.5$ at opposing MOD surface locations ($\theta = \pi$ and $\theta = 0$) in Experiment 2A. At this time, the MOD is near the $P_E$ peak and $H_S$ is the dominating factor in terms of $A_S$ production (right).

At $q_x$-minimum ($t \approx 8.0$)

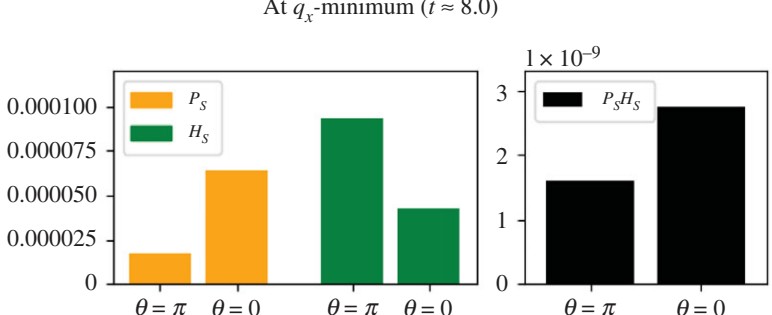

**Figure 11.** The concentration of $P_S$ and $H_S$ at $t \approx 9.5$, when the MOD approaches the peak of the $H_E$ gradient. Here $P_S$ is the dominating factor in terms of $A_S$ production.

plotted in figures 10 and 11 are not entirely symmetrical because the different adsorption rates of $H$ and $P$ are not the same.

## 3.3. Experiment 3: varying the adsorption rate of the precursor

In Experiment 2A, the MOD comes to rest at a point right of centre, i.e. a location higher in $P_E$ than of $H_E$. Stoichiometrically, the reaction requires and consumes $P$ and $H$ at the same rate, so what would cause this asymmetrical reaction of the MOD to its symmetrical environment? The answer to this question lies in the difference in the surface/environment exchange rates for $P$ and $H$ (see the $\chi$ column of table 1). We can confirm this by experimentally varying $\chi_P$, and observing how the MOD's final position changes as a result (plotted in figure 12). Here, we can see that lower values of $\chi_P$ cause the droplet to approach a final resting point with a higher $P$ to $H$ ratio, i.e. closer to the $P$-peak at $x = 1.5$. At intermediate values of $\chi_P$, the two resources are equally important and the MOD moves to a location between them, and for high values ($\chi_P \gtrsim 9$) the adsorption of $P$ appears to no longer be a significant constraint on the reaction rate, and the droplet moves to the peak of the other environmental resource, $H$.

# 4. Discussion

In the introduction, we suggested that MOD and related systems are capable of forms of self-preservation, and that some of these forms of self-preservation are 'viability-based', i.e. operate in response not to the environment, but to how well the system is managing to persist despite its intrinsic tendency to degrade. We then presented a series of experiments to demonstrate the dynamics of MOD motility in different contexts. The remainder of this paper considers the extent to which the demonstrated behaviours are self-preserving, and to what extent they are 'viability-based' and can thus adapt to changes in the system's existential needs.

In the first set of experiments, we simulated MOD that are incapable of 'refuelling,' i.e. replacing the fuel-like precursor ($P$) component that is required for continued motility. The motivation for this

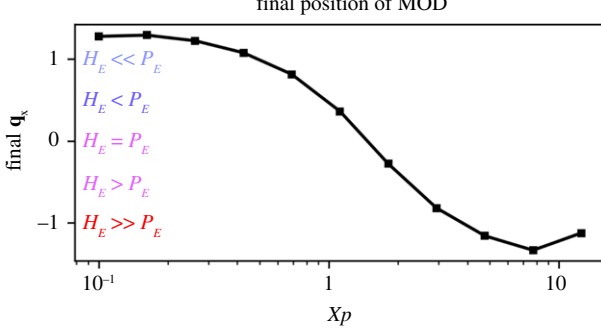

**Figure 12.** The final position of the droplet changes as a function of how rapidly $P$ is absorbed from the environment (described by parameter $\chi_P$. Peak $P_E$ values are found at $x = 1.5$; peak $H_E$ values are found at $x = -1.5$. Slower rates of $P$ adsorption cause the MOD to move towards a higher concentration of $P$. Final position was estimated by running simulation for 200 time units where the system was observed to have come to a steady state.

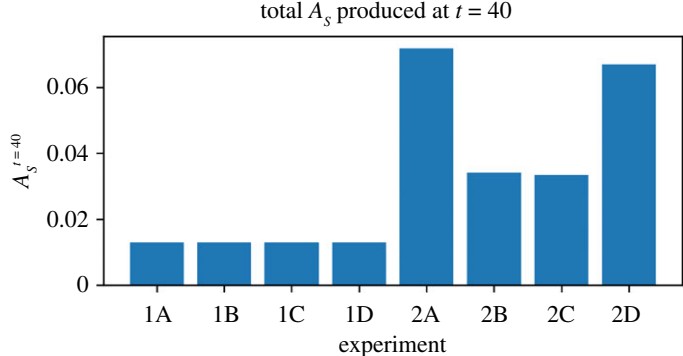

**Figure 13.** Total amount of $A_S$ produced at $t = 40$ for Experiments 1A–1D and 2A–2D. The first set of experiments all produce essentially the same quantity of $A_S$ as limited by the initial quantity of $P_S$. The second set of experiments produce more as they can replenish $P_S$ from their environment, with motile refuelling MOD (2A and 2D) finding more resources and producing more product than non-motile refuelling MOD (2B and 2C).

configuration was to recreate a system comparable to the MOD investigated in the laboratory. The reaction on its surface drove the MOD towards those local conditions which were more conducive to that very reaction. The reaction has certain requirements to take place (specifically, it needs $H$ and $P$), and it moves the system towards the only one of these reactants available in the environment ($H_E$). As such, we might consider the behaviour of the MOD to be responding to its own needs.

However, in the first set of our experiments, it may still be difficult to conclude that the behaviour of the droplet is in fact self-preserving. When the MOD is allowed to move (Experiments 1A and 1D), the reaction itself lasts less time than when the MOD is prevented from moving (Experiments 1B and 1C). By moving, the motility mechanism 'burns itself out'—it accelerates its own operation, and uses its finite store of $P$ more quickly than droplets which are prevented from seeking areas higher in $H$. The reaction lasts for less time (figure 7) and produces essentially the same quantity of product (figure 13).

The situation is different in the second series of experiments. When capable of replenishing $P$, the MOD moves towards both of the resources it needs for the reaction to occur, producing $A$ at a greater rate than droplets prevented from moving (figure 14). There are two observations here that can support the claim that the MOD's behaviour is a viability-based form of self-preservation, i.e. it adaptively responds to the changing needs of the system in a way that tends to prolong its existence. Before making those observations, it is worth first re-emphasizing a point from the introduction that it *is not the oil droplet that is self-preserving*. Oil droplets are passively stable systems that do not require a steady input of energy to persist. The reaction, the Marangoni flow and the motion of the droplet, on the other hand, are part of a precarious, dissipative structure that can only persist with the steady input of energy (here provided by $H$ and $P$ and their transformation into $A$). This collection of interdependent processes (henceforth 'the system') move themselves towards environmental

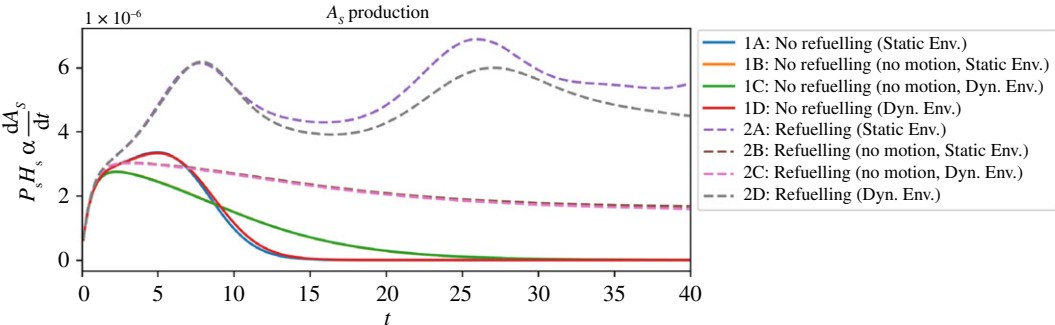

**Figure 14.** Relative rates of $A_S$ production for Experiments 1A–1D and 2A–2D as time passes.

conditions that are beneficial to their collective persistence, and as such this system is what we are arguing to accomplish self-preserving behaviour as defined in the introduction.

The first observation was highlighted in Experiment 3. When the environment was clamped, the MOD moved towards a point in between the two resources. The final resting point of the MOD varied in response to the needs of the system. We showed this by varying the rate of $P$ adsorption and observing that the final resting point varies in a way that adapts to the changed rate. Specifically, as $P$ becomes more difficult to adsorb (i.e. as $\chi_P$ decreases), it moves to a position that is higher in $P_E$ and lower in $H_E$. The system essentially 'selects' which environment to move to in response to needs that are dictated by its overall system dynamics, i.e. not just the chemical reaction (which requires $A$ and $P$ in equal amounts, but also by the other constraints that affect the availability of the two resources. There is of course no magic here—the behaviour, like the metabolism-dependent chemotaxis described in the introduction, is responding to the system's viability rather than directly to a environmental concentration, and so if something changes the way that the environment affects the system's viability, the behaviour can respond and adapt to that change as demonstrated in this model and elsewhere [19,20].

The second observation relates to the system's oscillatory behaviour, whereby it does not directly move to the steady state just described, but instead moves back and forth between the two resources in a damped oscillation. Our analysis of this motion showed how the behaviour (loosely speaking the decision to move to one resource or the other) depends upon the *current* state and needs of the system. When in an area high in the concentration of $H_E$, $P_E$ became the 'important', i.e. limiting factor in terms of the motility mechanism's persistence and the MOD moved towards that resource. And when in an area high in the concentration of $P_E$, $H_E$ became the limiting factor and the MOD responded in a viability-based and survival-prolonging manner by moving towards that resource. Here, at a shorter timescale, the behaviour of the MOD is again a response to its own dynamic (i.e. ever-changing) existential needs.

If these systems can accomplish viability-based behaviour, could there have been similar entities at the very earliest stages of life that regulated their interaction with their environment in a similar viability-based, adaptive and self-preserving manner? The role that self-preserving viability-based behaviours played in the earliest stages of life is not yet clear and further study is needed to understand the benefits and limitations of these kinds of behaviour, and under what conditions viability-based behaviour can emerge. In the meantime, it is also worth considering how viability-based behaviours could be employed by synthetic biologists who are working to building more robust and life like artefacts such as protocells. Currently, these systems require very specific laboratory conditions to persist, but perhaps if viability-based behaviours could be incorporated into such systems, they could play a greater role in their own persistence, regulating their own environment in response to their own viability dynamics in an adaptive and self-preserving manner that increases their robustness.

Data accessibility. I have made the code used in this project available for public download at https://github.com/matthew-egbert/motile_artificial_chemistry_droplets; for real-time visualization and interaction the project uses another package I created, also available at https://github.com/matthew-egbert/rvit and have been archived within the Zenodo repository: https://doi.org/10.5281/zenodo.5553448.
Competing interests. I declare I have no competing interests.

Funding. This research was funded by Royal Society Te Apārangi, Marsden Fund Te Pūtea Rangahau A Marsden grant number 17-UOA-196.

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
