## [Peer Review File · Royal Society Open Science]

Review History

RSOS-210534.R0 (Original submission)

Review form: Reviewer 1

Is the manuscript scientifically sound in its present form?

No

Are the interpretations and conclusions justified by the results?

Yes

Is the language acceptable?

Yes

Do you have any ethical concerns with this paper?

No

Have you any concerns about statistical analyses in this paper?

No

Recommendation?

Major revision is needed (please make suggestions in comments)

Comments to the Author(s)

It is an interesting article in an important emerging field. Suggested revisions will improve its chances of being read and cited (see Appendix A).

Review form: Reviewer 2**Is the manuscript scientifically sound in its present form?**

Yes

Are the interpretations and conclusions justified by the results?

Yes

Is the language acceptable?

Yes

Do you have any ethical concerns with this paper?

No

Have you any concerns about statistical analyses in this paper?

No

Recommendation?

Major revision is needed (please make suggestions in comments)

Comments to the Author(s)

The paper "Self-preserving Mechanisms in Motile Oil Droplets: A computational model of abiological self-preservation" is a very interesting contribution. However, I have definite problems with its rigorous understanding.

1. I have a much problem with understanding the main notion of the manuscript, namely "adaptive self-preservation". I think that a rigorous definition of this notion is definitely necessary. There exist a lot of self-propelling systems keeping for a long time their physical and chemical integrity, see for example:

B. P. Binks et al., Self-Propulsion of Liquid Marbles: Leidenfrost-like Levitation Driven by Marangoni Flow, . J. Phys. Chem. C 2015, 119, 18, 9910–9915.

Liquid marble, driven by the Marangoni soluto-capillarity flow, presented in this paper, keeps its shape and integrity even after numerous collisions with the Petri dish.. Is the system, presented in this paper regarded as demonstrating "adaptive self-preservation"? What is the criterion distinguishing "adaptive self-preservation"?

2. I did not like the analysis of hydrodynamics of the motion on the motile droplet, presented in the paper. It should start from the analysis of the dimensionless numbers, related to the reported physical process, namely: the Marangoni number, the Reynolds number, etc. This will clarify the mechanism of the motion.

3. Figures 1-2 need scale bars.

Decision letter (RSOS-210534.R0)

Dear Miss Egbert:

Title: Self-preserving Mechanisms in Motile Oil Droplets: A computational model of abiological self-preservation

Manuscript ID: RSOS-210534

The editor assigned to your manuscript has now received comments from reviewers. We would like you to revise your paper in accordance with the referee and Subject Editor suggestions which can be found below (not including confidential reports to the Editor). Please note this decision does not guarantee eventual acceptance.

Please submit your revised paper before 18-Jun-2021. Please note that the revision deadline will expire at 00.00am on this date. If we do not hear from you within this time then it will be assumed that the paper has been withdrawn. In exceptional circumstances, extensions may be possible if agreed with the Editorial Office in advance. We do not allow multiple rounds of revision so we urge you to make every effort to fully address all of the comments at this stage. If deemed necessary by the Editors, your manuscript will be sent back to one or more of the original reviewers for assessment. If the original reviewers are not available we may invite new reviewers.

On behalf of the Subject Editor Professor Anthony Stace and the Associate Editor Dr Debashree Ghosh.

RSC Associate Editor:

Comments to the Author:

The referees raise some valid scientific concerns as well as point towards the general lack of clarity in the paper. These issues need to be addressed by the authors along with a point-wise reply clarifying the changes made.

RSC Subject Editor:

Comments to the Author:

(There are no comments.)

Reviewers' Comments to Author:

Reviewer: 1

Comments to the Author(s)

It is an interesting article in an important emerging field. Suggested revisions will improve its chances of being read and cited.

Reviewer: 2

Comments to the Author(s)

The paper "Self-preserving Mechanisms in Motile Oil Droplets: A computational model of abiological self-preservation" is a very interesting contribution. However, I have definite problems with its rigorous understanding.

1. I have a much problem with understanding the main notion of the manuscript, namely "adaptive self-preservation". I think that a rigorous definition of this notion is definitely necessary. There exist a lot of self-propelling systems keeping for a long time their physical and chemical integrity, see for example:

B. P. Binks et al., Self-Propulsion of Liquid Marbles: Leidenfrost-like Levitation Driven by Marangoni Flow, . J. Phys. Chem. C 2015, 119, 18, [9910-9915](tel:9910-9915).

Liquid marble, driven by the Marangoni soluto-capillarity flow, presented in this paper, keeps its shape and integrity even after numerous collisions with the Petri dish.. Is the system, presented in this paper regarded as demonstrating "adaptive self-preservation"? What is the criterion distinguishing "adaptive self-preservation"?

2. I did not like the analysis of hydrodynamics of the motion on the motile droplet, presented in the paper. It should start from the analysis of the dimensionless numbers, related to the reported physical process, namely: the Marangoni number, the Reynolds number, etc. This will clarify the mechanism of the motion.

3. Figures 1-2 need scale bars.

Author's Response to Decision Letter for (RSOS-210534.R0)

See Appendix B.

RSOS-210534.R1 (Revision)

Review form: Reviewer 1

Is the manuscript scientifically sound in its present form?

Yes

Are the interpretations and conclusions justified by the results?

Yes

Is the language acceptable?

Yes

Do you have any ethical concerns with this paper?

No

Have you any concerns about statistical analyses in this paper?

No

Recommendation?

Accept as is

Comments to the Author(s)

I am glad to see this article in its current form. I hope this article will lead to more experiments in the study of motile droplets.

Review form: Reviewer 2

Is the manuscript scientifically sound in its present form?

Yes

Are the interpretations and conclusions justified by the results?

Yes

Is the language acceptable?

Yes

Do you have any ethical concerns with this paper?

No

Have you any concerns about statistical analyses in this paper?

No

Recommendation?

Accept as is

Comments to the Author(s)

After the revision the paper is publishable.

Decision letter (RSOS-210534.R1)

Dear Dr Egbert:

Title: Self-preserving Mechanisms in Motile Oil Droplets: A computational model of abiological self-preservation
Manuscript ID: RSOS-210534.R1

It is a pleasure to accept your manuscript in its current form for publication in Royal Society Open Science. The chemistry content of Royal Society Open Science is published in collaboration with the Royal Society of Chemistry.

On behalf of the Subject Editor Professor Anthony Stace and the Associate Editor Dr Debashree Ghosh.

RSC Associate Editor:
Comments to the Author:
(There are no comments.)

RSC Subject Editor:
Comments to the Author:
(There are no comments.)

Reviewer(s)' Comments to Author:

Reviewer: 2

Comments to the Author(s)

After the revision the paper is publishable.

Reviewer: 1

Comments to the Author(s)

I am glad to see this article in its current form. I hope this article will lead to more experiments in the study of motile droplets.

Self-preserving Mechanisms in Motile Oil Droplets: A computational model of abiological self-preservation

Matthew Egbert*

University of Auckland, Auckland, NZ (Dated: February 23, 2021)

Review:

The article has interesting content that is worthy of publication. It is, however, written in a manner that makes it hard to read and understand. They do not use standard terminology and notation that chemists and physicists use to describe such systems, thus making it difficult for the intended readers. It has superfluous mathematical detail (such as time $t \in \mathbb{R}$, $\mathbf{p} \in \mathbb{R}$ etc.; no need to specify that time and chemical concentration are real number!). I found it very hard to understand the terminology and the notation. I suggest the following changes be made in the article to improve its readability and suitability for publication in this journal. I do not recommend it for publication in the current form or with only minor changes.

- The authors need to use well-established terminology based in thermodynamics instead of creating a new and confusing terminology. After reading several times what the authors mean by "precarious systems", I realized they are talking about dissipative structures. The authors need to use the term "dissipative structures" in place of "precarious systems" for non-equilibrium self-organized systems. Dissipative structures have been known for nearly 50 years and there is vast amount scientific literature on this subject. They are the topic of the book by Nicolis and Prigogine cited by the authors (Ref [19] published in 1977) and discussed in textbooks on Thermodynamics (See for instance the widely used: Kondepudi & Prigogine: *Modern Thermodynamics: From heat engines to dissipative structures.*)

- Section II needs to be rewritten to make it understandable. It would greatly help if the authors include a figure with an oil drop (MOD) and the surrounding medium in which all the variables (\mathbf{q} , the position of oil drop's center, θ , the coordinate of the surface of the MOD, concentrations of A, P and H etc.) are clearly indicated. The mathematical generality in this section does not serve any useful purpose, it only obfuscates the simple chemistry of the system.

Basically, there are two sets of concentrations of A, P, and H, one set on surface the oil drop and one in the medium/environment in which the oil drop is moving. The two sets can be indicated with a sub- or superscript. For example, since A, P, and H are functions of the position, (x, y) , in the environment, they could be indicated as $A_E(x, y)$, $P_E(x, y)$ and $H_E(x, y)$, where the

subscript E stands for environment. Similarly for the concentrations on the surface of the drop can be denoted by $A_S(\mathbf{q}(t), \theta)$, in which the position of the drop \mathbf{q} is a function of time.

- Researchers in this field are familiar with diffusion-reaction equations of the form:

$$\frac{\partial P_E}{\partial t} = D_P \nabla^2 P_E + r(P_E, H_E, A_E)$$

in which D_P is the diffusion coefficient of P, etc. and $r(P_E, H_E, A_E)$ indicates the reactions that alter P_E , etc.

Similarly on the surface of the MOD, for the concentration of A_S , the diffusion-reaction equation is:

$$\frac{\partial A_S}{\partial t} = D_A \frac{\partial^2 A_S}{\partial \theta^2} + s(P_E, H_E, A_E)$$

in which D_A is the diffusion constant of A and s the reactions.

In fact, since there are only three concentrations, six equations can be explicitly written with the simple reaction rates in equations (11)-(13). I suggest using the usual notation of k for reaction rate constant instead of ϕ .

- On page 8, line 3: the word "containing" is repeated.
- What happens to A once it enters the drop is not clear. Does it accumulate in the drop or does it decompose and diffuse out of the drop?
- For the Marangoni flow, the author should clarify if equation (5) (for the effective surface tension Γ) is an assumption or if it is based on empirical data.
- Equation (8) for the reaction at the MOD surface needs clarification and better explanation of the terms. The parameter H_v that explains equilibrium for the exchange of a reactant, and $V(\mathbf{q}, \theta)$ needs better explanation. A figure will be very helpful.
- In equation (9) for the motion of the MOD, the term $(\theta + \pi/2)$ needs explanation. If $Gd\theta$ is the force on a surface element $d\theta$, isn't the total force the integral of $Gd\theta$, why the factor $(\theta + \pi/2)$?

These changes in section II will enable the reader to understand the results presented in section III. The results presented in section III are fine and interesting. In Fig 4, the Marangoni flow arrows should show the flow within the drop to complete the flow (as shown in Fig. 3).

With these changes I would recommend this article for publication.

Appendix B

I would like to sincerely thank both of the reviews for their helpful comments. I have responded to all of these comments in line below with the reviewer's comments given a gray background. I believe the manuscript is significantly improved thanks to the helpful comments given.

Sincerely,

Matthew Egbert

Reviewer #1

It is an interesting article in an important emerging field. Suggested revisions will improve its chances of being read and cited.

The article has interesting content that is worthy of publication. It is, however, written in a manner that makes it hard to read and understand. They do not use standard terminology and notation that chemists and physicists use to describe such systems, thus making it difficult for the intended readers. It has superfluous mathematical detail (such as time $t \in \mathbb{R}$, $p \in \mathbb{R}$ etc.; no need to specify that time and chemical concentration are real number!). I found it very hard understand the terminology and the notation. I suggest the following changes be made in the article to improve its readability and suitability for publication in this journal. I do not recommend it for publication in the current form or with only minor changes.

Thank you to this reviewer for the suggestions to improve the presentation of the model. I have responded to virtually all of the suggestions, making substantial changes to the manuscript as detailed inline below.

One of the very few suggestions that I didn't follow is the reviewer's point about the notation specifying the domain of time and different spaces in the paper—i.e. that $t \in \mathbb{R}$ and $\mathbf{q} \in \mathbb{R}^2$ (I never wrote in the paper that chemical concentration is a real number). In many models, time and position do not vary continuously, but discretely and in this paper there are two concepts of 'space' – the 2D 'environment' and the 1D, periodic 'surface' of the droplet. I thought it helpful to be explicit about all of these variables so as to make things clear for the reader. Perhaps it is a bit redundant, but I a bit of redundancy might help future readers confirm that their understanding of the paper is correct.

- Section II needs to be rewritten to make it understandable. It would greatly help if the authors include a figure with an oil drop (MOD) and the surrounding medium in which all the variables (q , the position of oil drop's center, q , the coordinate of the surface of the MOD, concentrations of A, P and H etc.) are clearly indicated.

These are good suggestions and as elaborated upon below, I have heavily rewritten Section II, removing the unnecessarily general description of the model, allowing for a much simpler and improved description of the model. I also added a diagram as suggested here (it is Figure 4 in the new version of the manuscript). Thank you to this reviewer for the helpful suggestions along these lines.

The mathematical generality in this section does not serve any useful purpose, it only obfuscates the simple chemistry of the system.

This is perhaps one of the most helpful comments of the whole review. The software I developed to simulate this system was designed to be easily modified so as to simulate other chemistries. That is what motivated my initial description of the model, but as the reviewer has pointed out, the generality is not relevant for the present paper and its inclusion is unhelpful.

Accordingly, I have removed the general description of the class of models that my program can simulate and replaced it with a set of differential equations that describe the precise system being simulated in this case, which I believe has significantly improved the manuscript. Thank you for this useful suggestion.

Basically, there are two sets of concentrations of A, P, and H, one set on surface the oil drop and one in the medium/environment in which the oil drop is moving. The two sets can be indicated with a sub- or superscript. For example, since A,P, and H are functions of the position, (x,y), in the environment, they could be indicated as $A_E(x,y)$, $P_E(x,y)$ and $H_E(x,y)$, where the subscript E stands for environment. Similarly for the concentrations on the surface of the drop can be denoted by $A_S(q(t),q)$, in which the position of the drop q is a function of time.

I have adpted the S and E subscripts as suggested for describing surface and environmental concentrations of reactants.

• Researchers in this field are familiar with diffusion-reaction equations of the form:

$$\frac{\partial P_E}{\partial t} = D_P \nabla^2 P_E + r(P_E, H_E, A_E)$$

in which D_P is the diffusion coefficient of P, etc. and $r(P_E, H_E, A_E)$ indicates the reactions that alter P_E , etc.

Similarly on the surface of the MOD, for the concentration of A_S , the diffusion-reaction equation is:

in which D_A is the diffusion constant of A and s the reactions.

$$\frac{\partial A_S}{\partial t} = D_A \frac{\partial^2 A_S}{\partial \theta^2} + s(P_E, H_E, A_E)$$

In fact, since there are only three concentrations, six equations can be explicitly written with the simple reaction rates in equations (11)-(13). I suggest using the usual notation of k for reaction rate constant instead of ϕ .

Agreed. I now describe the model using six differential Equations (Eqs. 1-5,12 in the manuscript) that have similar form to that suggested here. I have also changed ϕ to k throughout as suggested.

• What happens to A once it enters the drop is not clear. Does it accumulate in the drop or does it decompose and diffuse out of the drop?

I have added the following sentence to Section IIIA

Once created there is no degradation or removal of A_S and so it accumulates in the surface of the drop during the course of the simulation.

• For the Marangoni flow, the author should clarify if equation (5) (for the effective surface tension G) is an assumption or if it is based on empirical data.

To clarify this, I have modified the sentence that introduces Equation 5.

The interfacial tension, which varies locally over the surface of the droplet is calculated as the sum of the local concentration of the species, scaled by their surface tension constant.

has become

The interfacial tension, which varies locally over the surface of the droplet is assumed to be proportional to the sum of the local concentration of the species, scaled by their surface tension constant.

• Equation (8) for the reaction at the MOD surface needs clarification and better explanation of the terms. The parameter H_v that explains equilibrium for the exchange of a reactant, and $V(q,q)$ needs better explanation. A figure will be very helpful.

This is an entirely fair observation. I have rewritten the manuscript's explanation for the exchange between the MOD's surface and its environment. The new explanation is on pages 10–12, starting with the sentence "The last terms of Equations 1–2, and 4–5 describe exchange of reactants between the MOD surface and the environment." As suggested, I have included a Figure to help explain the weighting kernels (Fig. 5). I also renames ' H_v ' to ' α ' to avoid overloading the symbol ' H ' which is used to refer to one of the reactant species.

• In equation (9) for the motion of the MOD, the term $(\theta+\pi/2)$ needs explanation. If $Gd\theta$ is the force on a surface element dq , isn't the total force the integral of $Gd\theta$, why the factor $(\theta+\pi/2)$?

Flow on the surface is modelled as always tangential to the surface and the force due to the flux is similarly tangential. The purpose of the $(\theta+\pi/2)$ term is to translate the position of the flux on the surface of the droplet (θ) into a tangential vector $(\theta+\pi/2)$ which is then scaled by the flux velocity at that location. I have added the text in bold below to hopefully make this clearer for future readers. Thank you for this comment.

where $k_v = 2.5 \times 10^4$ is a constant that scales the amount of motion produced by the Marangoni flux **and the term $(\theta + \pi/2)$ specifies that the force applied is tangential to the surface.**

These changes in section II will enable the reader to understand the results presented in section III. The results presented in section III are fine and interesting. In Fig 4, the Marangoni flow arrows should show the flow within the drop to complete the flow (as shown in Fig. 3).

I hope that the changes I have made do indeed make the rest of the paper more intelligible. Thank you for your suggestions.

I have not added arrows to Fig. 4 (now Fig. 6) as there is no simulation in the model of internal fluid dynamics or replacement of material from the inside of the droplet.

The authors need to use well-established terminology based in thermodynamics instead of creating a new and confusing terminology. After reading several times what the authors mean by "precarious systems", I realized they are talking about dissipative structures. The authors need to use the term "dissipative structures" in place of "precarious systems" for non-equilibrium self-organized systems. Dissipative structures have been known for nearly 50 years and there is vast amount scientific literature on this subject. They are the topic of the book by Nicolis and Prigogine cited by the authors (Ref [19] published in 1977) and discussed in textbooks on Thermodynamics (See for instance the widely used: Kondepudi & Prigogine:

Thank you for this comment. The notion of precariousness and the concept of a dissipative structure are highly related. I didn't coin the term 'precariousness' (see e.g. [1]) but in any case, I agree that the paper is more clear if I stick with the more well known concept of 'dissipative structure' and have amended the text throughout accordingly.

[1] Di Paolo, Ezequiel A. "Extended Life." *Topoi* 28, no. 1 (March 1, 2009): 9--21.
<https://doi.org/10.1007/s11245-008-9042-3>.

Reviewer #2

The paper "Self-preserving Mechanisms in Motile Oil Droplets: A computational model of abiological self-preservation" is a very interesting contribution. However, I have definite problems with its rigorous understanding.

1. I have a much problem with understanding the main notion of the manuscript, namely "adaptive self-preservation". I think that a rigorous definition of this notion is definitely necessary. There exist a lot of self-propelling systems keeping for a long time their physical and chemical integrity, see for example: B. P. Binks et al., Self-Propulsion of Liquid Marbles: Leidenfrost-like Levitation Driven by Marangoni Flow, . *J. Phys. Chem. C* 2015, 119, 18, 9910-9915. Liquid marble, driven by the Marangoni soluto-capillarity flow, presented in this paper, keeps its shape and integrity even after numerous collisions with the Petri dish. Is the system, presented in this paper regarded as demonstrating "adaptive self-preservation"? What is the criterion distinguishing "adaptive self-preservation"?

Thank you for this useful comment. I have substantially reworked Section IA to include a more precise and formal description of what we mean by adaptive self-preservation including defining criteria.

2. I did not like the analysis of hydrodynamics of the motion on the motile droplet, presented in the paper. It should start from the analysis of the dimensionless numbers, related to the reported physical process, namely: the Marangoni number, the Reynolds number, etc. This will clarify the mechanism of the motion.

Other papers, e.g. "Self-maintained Movements of Droplets with Convection Flow" https://link.springer.com/chapter/10.1007/978-3-540-76931-6_16 have evaluated in detail the fluid dynamics involved in the mechanism of motility. The focus of the present paper is different: it aims to evaluate the relationship between the motility and its effect upon the reaction that drives the motility operation. As such it is a "higher-level" model, i.e. not simulated using navier-stokes equations, but building a more 'coarse grained' model that takes the motility as given (supported by previous work such as the aforementioned paper). Accordingly we haven't included analysis of the mentioned dimensionless numbers in this paper.

To make more clear this motivation for the paper, we amended the abstract to read as follows:

Recent empirical work has characterized *motile oil-droplets*—small, self-propelled oil-droplets whose active surface-chemistry moves them through their aqueous environment. Previous work has evaluated in detail the fluid dynamics involved in the mechanism of motility. This paper introduces a new computational model that is used to evaluate the behaviour of these droplets *as a form of adaptive self-preservation*, whereby (i) the mechanism of motility causes motion toward the

conditions conducive to that mechanism's persistence; and (ii) this form of 'viability-sensitive' self-preservation automatically adapts to compensate when the motility mechanism's ideal operating conditions change. The model simulates a motile oil droplet as a disc that moves through a 2D spatial environment containing diffusing chemicals. The concentration of reactants on its surface change by way of chemical reactions, diffusion, Marangoni flow (the equilibration of surface tension), and exchange with the droplet's local environment. Droplet motility is a byproduct of Marangoni flow, similar to the motion-producing mechanism observed in the lab.

In addition, the following text was added to the introduction, to include the reference to previous work.

The main body of the paper presents a computational model of motile oil droplets similar to systems that have been fabricated and investigated in the laboratory—e. g. [13, 15]. **Previous computational models have investigated the fluid dynamics underlying the motility of these droplets [17].** We use our model to investigate how the chemical reactions taking place on the surface of droplet cause the droplet to move toward conditions that facilitate or extend the life of those very same reactions—a basic form of self-preservation.

3. Figures 1-2 need scale bars.

I added scale bars to Figure 2 and amended the caption of Figure 1 to inform the reader of the scale of the image. The caption now reads:

A ramified charge-transportation network (RCTN). When conductive chromium spheres are partially submerged in oil and subjected to a high voltage potential, they self-organize into dendritic structures such as that shown here. In this experiment, the spheres are 4mm in diameter.

I also replaced Figure 2 with an equivalent image that is licensed under a Creative Commons Attribution-NonCommercial 4.0 International License.